# Could Vulnerability to Motion Sickness and Chronic Pain Coexist within a Sensorimotor Phenotype? Insights from over 500 Pre-Pain Motion Sickness Reports

**DOI:** 10.3390/brainsci13071063

**Published:** 2023-07-12

**Authors:** Daniel Simon Harvie

**Affiliations:** IIMPACT in Health, Allied Health and Human Performance Unit, University of South Australia, Adelaide, SA 5000, Australia; daniel.harvie@unisa.edu.au

**Keywords:** chronic pain, multisensory integration, motion sickness, sensorimotor incongruence

## Abstract

Background: The sensorimotor incongruence theory proposes that certain instances of pain result from conflicts in the brain’s sensorimotor networks. Indeed, injuries may cause abnormalities in afferent and cortical signaling resulting in such conflicts. Motion sickness also occurs in instances of incongruent sensorimotor data. It is possible that a sensory processing phenotype exists that predisposes people to both conditions. Aim: The aim of this study was to investigate whether participants with chronic pain recall greater susceptibility to motion sickness before chronic pain onset. Method: Data were collected via an online LimeSurvey. A self-report tendency toward motion sickness was measured using the Motion Sickness Susceptibility Questionnaire. Group differences were analysed using analysis of covariance methods. Results: 530 patients (low back pain, *n* = 198; neck pain, *n* = 59; whiplash-associated disorder, *n* = 72; fibromyalgia syndrome, *n* = 114; Migraine, *n* = 41) and 165 pain-free controls were surveyed. ANCOVA analysis, using sex and anxiety as covariates, suggested that childhood motion sickness susceptibility scores differed by group (F = 2.55 (6, 615), *p* = 0.019, ($\eta_{p}^{2}$) = 0.024). Planned comparisons, with corrected *p*-values, suggested that childhood motion sickness was not statistically greater for low back pain, rheumatoid arthritis, migraine, neck pain or whiplash-associated disorder (*p*s > 0.05), although scores were on average 27%, 42%, 47%, 48% and 58% higher, respectively. Childhood susceptibility was statistically higher in people with FMS (*p* = 0.018), with scores on average 83% higher than controls. ANCOVA analysis, using sex and anxiety as covariates, suggested that adult motion sickness susceptibility scores did not differ by group (F = 1.86 (6, 613), *p* = 0.086), although average scores were, on average, at least 33% higher in persistent pain groups. Conclusions: According to retrospective reporting, greater susceptibility to motion sickness appears to pre-date persistent pain in some conditions. This supports the possibility that motion sickness and chronic pain may, in some cases, have overlapping mechanisms related to the handling of incongruent sensorimotor data.

## 1. Introduction

Chronic pain affects one in five people and stands as the greatest cause of years lived with disability [1]. Today much is still unknown about why pain can persist beyond injury healing, and several theories attempting to explain it remain largely unresolved [2,3,4,5]. The sensorimotor incongruence theory of pain [4] suggests that some cases may be a response to persistent conflicts within the brain’s sensorimotor networks, although the reason this would cause pain is not known. Nonetheless, it is certainly true that injuries may cause persistent abnormalities in afferent signaling and/or adaptations in sensorimotor networks that disturb sensory processing and induce a state of incongruence between predictions regarding motor-related sensory feedback and actual sensory feedback [4,6].

Interestingly, another unpleasant phenomenon—namely motion sickness—also occurs in response to incongruent sensorimotor data [7]. Here, it is posited that sickness is a protective response to a state of sensorimotor incongruence that mimics a neurotoxic state that would occur following poison ingestion [8]. The motion sickness phenomenon lends support to the biological relevance of states of sensorimotor incongruence and further suggests that we may have evolutionarily prepared protective responses that may be triggered in response. Given the similarity between the proposed processes, it is not surprising that motion sickness was used as an analogy when originally describing the sensorimotor incongruence theory of persistent pain [7]. Importantly, since it appears that some individuals are more susceptible to these two unpleasant states than others, it is possible that a sensory processing phenotype exists that predisposes people to one or both.

### 1.1. Overlapping Processes of Motion Sickness and Pain

Our brain predicts the sensory outcome of our movements based on motor commands [9]. This prediction is compared to actual sensory feedback, allowing for adjustments to be made during movement [9]. Some types of persistent pain are thought to arise from conflicting sensory and motor signals, which induce a state of so-called sensorimotor incongruence (SMI) [4]. SMI was first applied to the pain context in attempt to explain phantom limb pain, where motor signals from the missing limb cannot be resolved with congruent sensory feedback [4]. The principle behind mirror therapy is consistent with SMI theory since visual feedback provided by the reflected intact limb aligns with the intended movement, thereby restoring sensorimotor congruence. However, it seems that, at best, mirror therapy is only effective in subgroups of phantom limb pain [10], with meta-analyses suggesting an overall analgesic effect of just 1.3/10 (on a 10-point Visual Analogue Pain Rating Scale) [11]. Notably, attempts to cause pain using experimentally induced SMI in healthy controls have failed [12]. However, interest remains in how more ecologically valid and persistent states of SMI may contribute to maintaining persistent pain states [6]. For example, inducing sensory conflict has been shown to increase clinical limb pain and pain sensitivity in people with pathological hand pain and increase reports of pain in some people with fibromyalgia syndrome [13,14].

### 1.2. Neural Basis of Motion Sickness and Pain

Neuroimaging studies confirm that pain is associated with activity in a wide range of brain regions with diverse functions [15]. Observation of brain regions associated with both pain and motion sickness may assist in identifying plausible common mechanisms. The cerebellum is one such region [16,17]. Traditionally, the cerebellum was thought to be involved only in motor coordination and motor learning [18]. However, recent studies have shown that it plays an important role in a range of non-motor functions [19]. Indeed, the cerebellum is highly interconnected within the brain and receives input from various regions, including those related to motor, vestibular, somatosensory, cognition, emotion and pain [20,21].

One of the key functions of the cerebellum is to compare sensory data with motor commands to detect discrepancies from expected feedback and aid the correction and coordination of movement [18]. As a result, the cerebellum plays a key role in detecting the sensory conflicts—for example, between predicted vestibular, kinesthetic feedback and actual feedback—that lead to motion sickness. In the case of pain, the cerebellum receives input from primary nociceptive afferents and has bidirectional connections with the dorsolateral prefrontal cortex and other brain regions that modulate sensitivity in pain-related pathways, making it well-positioned to influence pain [17]. Moreover, cerebellar dysfunction has been linked to various pain disorders, such as migraine headaches [22], where sickness symptoms are also common [23]. As such, the cerebellum is a plausible locus for overlapping mechanisms of pain and motion sickness.

### 1.3. The Sensorimotor Conflict Phenotype Hypothesis

If a sensory processing phenotype exists that increases susceptibility to both chronic pain and motion sickness, then populations with chronic pain would likely have a higher prevalence of motion sickness susceptibility. Additionally, chronic pain conditions where sensorimotor incongruence is more relevant may show higher rates of motion sickness susceptibility than other chronic pain conditions. Therefore, our aim was to assess whether various groups of people with chronic pain report greater motion sickness susceptibility compared to those without pain.

## 2. Methods

### 2.1. Design

Data were collected using an online survey on the LimeSurvey platform [24]. The survey, which took approximately 20 min to complete, was conducted between September 2015 and July 2016. Links to the survey were advertised using a broad national and international strategy encompassing social media and organizational websites related to Arthritis Australia, and the Body in Mind research group. Ethics were approved by Griffith University Human Research Ethics Committee (HMR/01/15HREC). 

### 2.2. Subjects

Participants who had experienced pain for longer than three months were included in the study. Participants who currently had no pain and had no history of chronic pain were also included as a control group. Anyone whose pain started before the age of 12 was excluded, such that pre-pain childhood motion sickness susceptibility scores could be garnered. The test sample consisted of individuals who reported themselves at the beginning of the survey as having persistent low back pain, migraine, neck pain, whiplash-associated disorder, fibromyalgia, or rheumatoid arthritis. Rheumatoid arthritis is a persistent pain condition with a clear tissue-based inflammatory mechanism, in contrast to other conditions where local tissue factors are likely less dominant and central nervous system contributors are present. Therefore, rheumatoid arthritis was included as an additional control group to help control for potential confounders, such as current pain, medication and recall bias. That is, rheumatoid arthritis provides a distinct example of a pain condition driven by tissue inflammation, which may help to isolate the specific effects of the central factors under investigation.

### 2.3. Outcomes

The survey used the Motion Sickness Susceptibility Questionnaire (MSSQ-Short [25]), which requires participants to self-report their motion sickness before the age of 12, and their motion sickness experience as an adult over the last 10 years. The MSSQ-Short asks participants to rate how often they felt nauseated during nine different transport- and play-related activities on a scale of 0 = never felt sick, to 3 = frequently felt sick, with activities not experienced left blank. A score is calculated that is equal to the (total sickness score child) × (9)/(9 − number of types not experienced as a child), producing a maximum score of 27, achievable regardless of the number of items experienced. This is the most used questionnaire in motion sickness research and has good reliability (correlation among repeated measures of r = 0.90 for childhood scores and *r* = 0.68 for adult scores) [25]. Moreover, it has a justifiable compromise between length and validity—it has a correlation of 0.94 with the MSSQ-Long and predicts nausea during provocative translational oscillations of *r* = 0.74 [25]. Additionally, participants provided self-reported descriptive information on the duration of their chronic pain, their level of disability (Pain Disability Index [26]), pain duration, and pain severity (NRS = Numerical Pain Rating Scale). This information was collected online through multiple-choice questions and numerical rating scales. Generalised Anxiety Disorder (GAD7 [27]) data was also collected as a descriptor of psychological status due to its potential association with pain and motion sickness.

### 2.4. Analysis

Statistical analysis of the data was performed using IBM SPSS v29.0, Chicago, IL, USA. The primary outcome was retrospective reports of motion sickness susceptibility during childhood, before the onset of persistent pain. This was because current adult reports of motion sickness were considered likely to be confounded by persistent pain and related factors such as medications and sleep disturbances. Differences and influence of demographic characteristics were assessed. This led to the use of sex and anxiety as covariates in the analysis. Separate one-way analysis of covariance (ANCOVA) was performed to assess differences in motion sickness among the seven groups (control, low back pain, neck pain, whiplash-associated disorder, rheumatoid arthritis, fibromyalgia syndrome, and migraine) for adult and childhood scores. Planned comparisons among groups were then performed separately to identify groups that were different to the controls. Holm–Bonferroni *p*-value adjustments were made to account for multiple comparisons [28]. Effect sizes ($\eta_{p}^{2}$) were calculated, and descriptive statistics were provided to clarify the magnitude of any apparent group differences. Equal group sizes are not an assumption for ANOVA or pairwise testing in SPSS [29]; however, imbalances in group sizes were noted.

## 3. Results

### 3.1. Descriptive Data

The total number of participants was 695; 530 participants reported currently experiencing chronic pain, whereas 165 participants were pain-free controls. A breakdown of chronic pain subgroups is displayed in Table 1 and Table 2.

### 3.2. Group Differences in Characteristic

It was noted that pain-free control subjects were, on average, eight years younger than the mean across the sample. To confirm if the age of participants could meaningfully impact motion sickness susceptibility reports, we compared scores in those controls aged above the mean to those under. The scores were almost identical between age groups (Mean difference (MD), Standard Error (SE) Childhood = −0.1 (1.0); Adult = 0.5 (SE0.9)), and independent *t*-tests did not find evidence of an age-related effect for either child (t = −0.096 (158), *p* = 0.923) or adult (t = −0.058 (158), *p* = 0.954) reports of motion sickness susceptibility.

Female representation was higher in both patient and control groups, and the extent of this imbalance differed across groups. While sex differences in motion sickness susceptibility did not reach statistical significance for childhood scores (t = 1.79 (158), *p* = 0.75, MD (SE) = 1.8 (1.0), they did for current adult scores (t = 2.29 (158), *p* = 0.023, MD (SE) = 2.1 (0.9). To account for its potential impact on the analysis, sex was included as a covariate.

Anxiety scores were also notably higher across all patient groups relative to controls (Table 1). Anxiety is known to have a weak correlation with motion sickness susceptibility [30], and data here were consistent with that finding for childhood (Pearson’s *r* = 0.197, *p* < 0.001) and adult motion sickness (*r* = 0.225, *p* < 0.001). Observation of *r*^2^ values estimated a shared variance of just 4–5%. Nonetheless, given the significant differences in anxiety scores across the groups, anxiety (GAD7) scores were included as a second covariate.

### 3.3. Primary Analysis

ANCOVA analysis suggested that childhood motion sickness susceptibility scores differed by group (F = 2.55 (6, 615), *p* = 0.019, ($\eta_{p}^{2}$) = 0.024) (Figure 1). Holm–Bonferroni corrected planned comparisons suggested that childhood motion sickness was not greater than controls for low back pain, rheumatoid arthritis, migraine, neck pain and whiplash-associated disorder (*p* = 0.95, 0.95, 0.60, 0.54, 0.07, respectively) although scores were on average 27%, 42%, 47%, 48% and 58% higher (Figure 1, Table 2). Childhood (pre-chronic pain) reports, however, were statistically higher in people with FMS (*p* = 0.018), with scores on average 83% higher than controls.

### 3.4. Secondary Analysis

ANCOVA analysis with sex and anxiety as covariates did not support statistically significant differences between groups in adult motion sickness susceptibility scores (F = 1.86 (6, 613), *p* = 0.086, ($\eta_{p}^{2}$) = 0.018) (Figure 1). We nonetheless proceeded with planned Holm–Bonferroni corrected comparisons, which also did not support statistical differences in adult motion sickness for rheumatoid arthritis, low back pain, neck pain, and whiplash-associated disorder (*p* = 0.74, 0.74, 0.234, 0.228, 0.170, respectively) although scores were on average 33%, 37%, 63%, 65% and 81% higher (Figure 1). Migraine also failed to reach a statistically supported difference (*p* = 0.054), although scores were nearly double (on average 94% higher).

## 4. Discussion

This study was based on the proposition that a sensory processing phenotype may exist relating to the management of conflicting sensorimotor data and that this may predispose individuals to both motion sickness susceptibility and chronic pain. Our hypothesis postulated that participants with chronic pain would exhibit a higher susceptibility to motion sickness compared to controls. Furthermore, we hypothesised that this heightened susceptibility would be evident in childhood, preceding the onset of chronic pain. Our hypothesis was supported for certain chronic pain conditions but not others. Specifically, a significant association was found between greater sensitivity to motion sickness before the onset of chronic pain in individuals with fibromyalgia syndrome (FMS). While the study design does not allow for definitive conclusions, this finding aligns with the possibility that a sensory processing phenotype exists that predisposes individuals to both motion sickness and some persistent pain conditions. Importantly, all conditions reported at least a 27% higher susceptibility score compared to controls, indicating the possibility of additional group differences that our study may not have had sufficient power to detect due to the large variability in scores.

While the study was cross-sectional, we requested participants to report their motion sickness susceptibility during childhood (<12 years old). To prevent potential interactions between pain and motion sickness in adult data, we excluded participants whose pain began before the age of 12. Therefore, the current data support the notion that increased motion sickness susceptibility predates the onset of chronic pain in certain conditions. However, we cannot completely rule out the possibility of a pain-related reporting bias. Although we included rheumatoid arthritis (RA) as a control group for this reason, we later discovered that RA can impact the vestibular system [31,32]. Given that RA could potentially influence motion sickness susceptibility, even before other clinical symptoms manifest, it becomes an uncertain control group. Nevertheless, the elevated childhood reports of motion sickness susceptibility in individuals with RA, though not statistically significant, warrant caution regarding the potential confounding effect of pain-related reporting bias in this study. However, it seems unlikely that a reporting bias alone would explain an 83% higher incidence of childhood motion sickness susceptibility (pre-pain onset) compared to controls.

Interestingly, individuals with fibromyalgia syndrome (FMS) reported the highest prevalence of childhood motion sickness. This observation holds significance because, among the conditions included in the study, FMS is sensitizat as being closely associated with central nervous system processes such as central sensitization compared to other persistent pain conditions [33,34]. Notably, the evidence supporting central involvement in FMS included alterations in cerebellar activity and connectivity [35,36]. Further, there is perhaps the most evidence supporting the relevance of sensorimotor Incongruence for FMS than for any other condition [6,13,14].

Current adult motion sickness susceptibility scores appeared higher in patient groups, particularly in individuals with migraine and fibromyalgia syndrome (FMS) (81% and 94% higher, respectively). However, upon including anxiety levels and sex in the analysis and correcting for multiple comparisons, there was no statistically meaningful evidence of group differences. This might also reflect the large variability in scores that might have been better accounted for by a larger sample. As such, we cannot rule out the possibility that increased motion sickness susceptibility may be a feature of a number of chronic pain conditions. Indeed, previous data has already reported the issue of motion sickness susceptibility in migraine sufferers [23].

Interestingly, migraine was the only condition in which adult motion sickness scores surpassed childhood reports. This finding is not surprising, considering that symptoms of motion sickness overlap with those of migraines, including headache, upper abdominal sensations, dizziness, and sleepiness [37]. Notably, a statistical interaction between anxiety and adult motion sickness was observed, unlike in childhood reports. That is, when anxiety was included as a covariate, the overall effect of the group was no longer significant for adult scores. Anxiety is known to have a two-way interaction with persistent pain [38], and its elevated presence among individuals with chronic pain may contribute to greater susceptibility to motion sickness. However, the shared variance accounted for by anxiety was only 5%, suggesting that other factors are likely involved in conditions where clear differences were observed.

### Strengths, Limitations and Future Directions

A key strength of the study was its inclusion of 530 people with chronic pain and 165 controls.

The main limitation of this study was its reliance on retrospective reports of childhood (pre-pain) motion sickness, which are susceptible to bias. In other words, the retrospective reports served as a proxy for what would ideally be a prospective design. However, conducting a prospective study to test such a speculative hypothesis would require significant resources and time, which may not be justified or ethical. While this study suggests the value of further exploring this hypothesis, we do not recommend a dedicated prospective study at this time. Rather, incorporating questionnaires related to sensory processing outcomes, such as motion sickness susceptibility, interoceptive awareness, somatosensory amplification, and sensory acuity, as part of larger prospective studies investigating predictors of persistent pain would, in our view, be justified. It is worth also noting that the idea that sensorimotor conflict may be present in people with persistent pain remains theoretical, although justifiable [6]. Moreover, methods to identify conclusively whether someone is in such a state do not exist. This presents a significant limitation to the current study and a challenge to future progress.

## 5. Conclusions

Based on retrospective reports, it seems that increased susceptibility to motion sickness predates the onset of persistent pain in fibromyalgia syndrome. This suggests the potential existence of shared mechanisms between motion sickness and chronic pain, particularly in relation to the processing of conflicting sensorimotor information. These findings align with the sensorimotor incongruence theory of persistent pain, which remains controversial.

## Figures and Tables

**Figure 1 brainsci-13-01063-f001:**
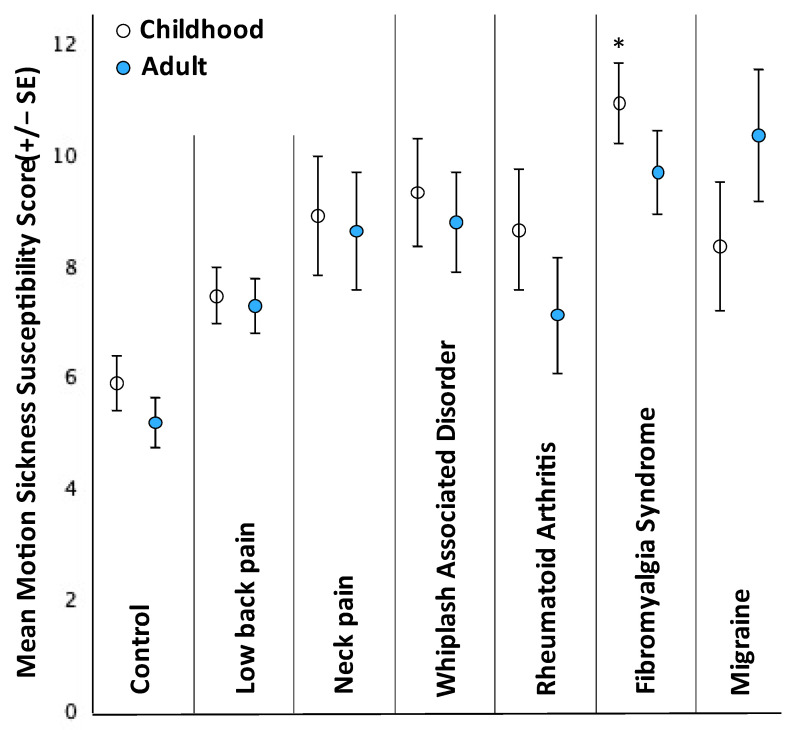
Adult and Childhood Motion Sickness Susceptibility scores for each group. * Represents statistically significant difference to control (*p* < 0.05).

**Table 1 brainsci-13-01063-t001:** Participant descriptive data.

	Controls	LBP	NP	WAD	RA	FMS	Migraine	Total
*n*	165	198	59	72	46	114	41	695
Males	62	49	16	10	5	5	3	150
Females	103	149	43	62	41	109	38	545
Age Mean (SD)	38 (12)	48 (12)	51 (12)	45 (13)	49 (14)	47 (13)	45 (12)	46 (13)

LBP = Lower back pain, NP = Neck pain, WAD = Whiplash-associated disorder, RA = Rheumatoid arthritis, FMS = Fibromyalgia syndrome.

**Table 2 brainsci-13-01063-t002:** Participant clinical, psychological, and motion sickness susceptibility data.

	Controls	LBP	NP	WAD	RA	FMS	Migraine	Total
Duration (years) Mean (SD)	N/A	13.1 (10.8)	13.7 (11.1)	12.0 (9.6)	9.8 (11.4)	12.9 (9.9)	20.7 (12.7)	13.3 (10.9)
Av. Pain Intensity 0–10	N/A	5.8 (2.1)	5.7 (2.0)	5.4 (2.1)	5.9 (1.7)	6.7 (1.8)	4.5 (2.8)	5.8 (2.1)
Worst Pain Intensity 0–10	N/A	7.8 (2.2)	7.4 (2.4)	7.4 (2.4)	8.4 (1.6)	8.7 (1.7)	6.7 (3.4)	7.9 (2.3)
GAD7 score Mean (SD)	3.0 (3.5)	7.0 (5.7)	6.0 (5.2)	6.1 (4.5)	7.0 (5.6)	8.3 (5.4)	6.3 (5.1)	6.0 (5.3)
PDI score Mean (SD)	N/A	39.6 (19.3)	39.3 (18.2)	34.0 (20.0)	45.7 (14.7)	49.0 (15.4)	37.4 (18.7)	41.2 (18.7)
MSSQ Child Mean (SD)	6.0 (6.2)	7.6 (7.3)	8.9 (8.2)	9.5 (8.2)	8.8 (7.5)	11.0 (7.8)	8.5 (7.3)	8.2 (7.5)
MSSQ Adult Mean (SD)	5.4 (5.7)	7.4 (6.9)	8.8 (8.2)	8.9 (7.6)	7.2 (7.1)	9.8 (8.0)	10.5 (7.4)	7.8 (7.2)

LBP = Lower back pain, NP = Neck pain, WAD = Whiplash-associated disorder, RA = Rheumatoid arthritis, FMS = Fibromyalgia syndrome, GAD7 (Generalised Anxiety Disorder Scale), PDI (Pain Disability Index).

## Data Availability

Open data is available at https://doi.org/10.7910/DVN/QO93HK.

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
