# Peer review of "Could Vulnerability to Motion Sickness and Chronic Pain Coexist within a Sensorimotor Phenotype? Insights from over 500 Pre-Pain Motion Sickness Reports"

_brainsci, 2023, doi:10.3390/brainsci13071063_

Round 1

Reviewer 1 Report

Dear editor,

Thank you for the invitation to review the manuscript titled “Could Vulnerability to Motion Sickness and Chronic Pain Co-2 exist within a Sensorimotor Phenotype? Insights from over 500 3 pre-pain motion sickness reports”.

The author presented an analysis of retrospective accounts of motion sickness in childhood and adulthood from a relatively large sample of participants, most of whom have chronic pain. The manuscript is framed in the context of the sensorimotor theory of pain, which seems highly appropriate for the special issue “Effect of Altered Sensory and Cognitive Processing on Sensorimotor Integration”. The main finding is that retrospective self-reports of childhood motion sickness were higher for people with fibromyalgia and whiplash associated disorder than for pain-free controls. Overall I think the manuscript is well written and constitutes a meaningful contribution to the field. However, there are several issues that need to be addressed.

I have two main concerns with the manuscript.

1)      I am not convinced that removing anxiety as a covariate is appropriate. The author states that anxiety only accounted for 5% of the variance of adult motion sickness scores. I would be interested to know what percentage Sex accounted for, as it was considered an appropriate covariate to include in the ANCOVA. Without a preregistered analysis plan, it is difficult to justify this choice and make a convincing case that it is not p-hacking. To be perfectly clear, I am not accusing the author of the latter, and I appreciate that preregistration was not commonplace in 2015. Rather, I think that the author needs to make a compelling argument for the removal of anxiety as a covariate so that the results are not dismissed as p-hacking. Or, in my opinion, a more appropriate alternative would be to include anxiety as a covariate and report on only these analyses.  

2)      The manuscript is missing key literature (e.g. Knudsen & Drummond, 2015, European Journal of Pain), and the other relevant papers by the same author that are cited are discussed in little detail. In addition to their interesting findings, these studies provide a direct measure of susceptibility to motion sickness, which is fundamental to the rational for the current manuscript. I think it is misleading to downplay their role in your manuscript. I would strongly recommend that you include a consideration of these studies the introduction and the discussion sections of your manuscript.

In addition to my two major concerns, I have a few minor comments that I believe would improve the manuscript. Below, I have organised these comments by section and line number.

Abstract

Line 22: The “ACOVA suggested”, it did not confirm.

Line 26: Lower than what?

Introduction

I think it is also appropriate to mention that Harris used motion sickness as an analogy when proposing his theory.

Line 64: The efficacy of mirror therapy is overstated. There are several meta-analyses that conclude that the evidence is not sufficiently strong to recommend mirror therapy as a treatment for phantom limb pain (Thieme et al., 2016, The Journal of Pain; Wittkopf et al., 2017, European Journal of Pain). Most of the evidence in favour of mirror therapy for phantom limb pain is based on case studies. I think more detail is needed in this sentence for it to be accurate.

Method

Line 109: How did participants “self-identify” with chronic pain conditions?

I think it is more appropriate to mention that you excluded any participants who had pain below the age of 12 years in the method section rather than leaving it for the discussion (i.e. line 225.

Line 138: Would you not expect pain and medication to induce a recall bias too?

Line 149: There is a reference missing to support the statement about the assumptions of ANOVA.

Results

Line 169: Strictly speaking, the analysis did not find evidence of a difference (i.e. it did not find evidence of no difference). This type of frequentist statistics cannot confirm a null-hypothesis, which would require a test of equivalence (e.g. for a frequentist alternative see Lakens, 2017, Social Psychology and Personality Science).

Line 184: What was the R2 for Gender in your ANCOVA?

Line 188: Greater than what?

Line 196: Figure 1: Not clear what the asterisks refer to. Presumably it is a significance level of the child-adult comparison per group. Either way, this information should be stated explicitly in the caption. It would also be useful to note what the maximum possible score is on the MSSQ-Short.

Discussion

Line 209: I do not think it is accurate to say that this is the first study to propose a sensory processing phenotype. The theory you are testing used motion sickness as an analogy, several previous studies have looked at motion sickness and pain, and there is an entire field dedicated to finding sensory phenotypes of pain (i.e. quantitative sensory testing). So I think some moderation here is warranted.

Line 245: In support of your argument, there is also greater evidence that sensorimotor incongruence is related to pain for people with FMS than for other conditions (see Vittersø et al. 2022, Neuroscience & Biobehavioral Reviews).

Line 286: The review article that you cite (i.e. Vittersø et al. 2022, Neuroscience & Biobehavioral Reviews) contains about 350 references to predominantly empirical studies that are relevant to understanding the sensorimotor theory of pain. So I do not think it is entirely accurate to say that the theory “lacks substantial empirical support”. I would suggest a more nuanced assessment of its empirical backing, although I agree that the evidence base is mixed. 

Line 230. Typo “.Although” – no space after full stop.

Line 238: Typo. Sentence ends with a quotation mark.

Line 240: Reported not exhibited.

Line 252: Unclear sentence. Are you saying that pre-pain scores indicate a pre-existing predisposition to pain? I think this sentence needs to be simplified.

Line 278: Typo. “Top of form; Bottom of form”.

Author Response

Reviewer 1

The author presented an analysis of retrospective accounts of motion sickness in childhood and adulthood from a relatively large sample of participants, most of whom have chronic pain. The manuscript is framed in the context of the sensorimotor theory of pain, which seems highly appropriate for the special issue “Effect of Altered Sensory and Cognitive Processing on Sensorimotor Integration”. The main finding is that retrospective self-reports of childhood motion sickness were higher for people with fibromyalgia and whiplash associated disorder than for pain-free controls. Overall I think the manuscript is well written and constitutes a meaningful contribution to the field. However, there are several issues that need to be addressed.

Author Response

Thank you for this summary and for your time reviewing the manuscript.

Reviewer 1

I have two main concerns with the manuscript.

  • I am not convinced that removing anxiety as a covariate is appropriate. The author states that anxiety only accounted for 5% of the variance of adult motion sickness scores. I would be interested to know what percentage Sex accounted for, as it was considered an appropriate covariate to include in the ANCOVA. Without a preregistered analysis plan, it is difficult to justify this choice and make a convincing case that it is not p-hacking. To be perfectly clear, I am not accusing the author of the latter, and I appreciate that preregistration was not commonplace in 2015. Rather, I think that the author needs to make a compelling argument for the removal of anxiety as a covariate so that the results are not dismissed as p-hacking. Or, in my opinion, a more appropriate alternative would be to include anxiety as a covariate and report on only these analyses.  

Author Response to ‘’major concern 1”

Thank you for the suggestion to contrast the variance explained by anxiety, to that explained by sex. Given that there was apparently similar shared variance (both R2 show approximately 4%) I could no longer justify omitting anxiety as a co-variate. I have re-run the analyses with gender and anxiety as covariates. The results are generally similar, but now Fibromyalgia Syndrome is the only group to show significant differences to the pain-free controls (Whiplash Associated Disorder was still significantly at p=0.14, however this did not survive p-value corrections as it did in the initial analysis). The results and discussion are now updated in line with this.

  • The manuscript is missing key literature (e.g. Knudsen & Drummond, 2015, European Journal of Pain), and the other relevant papers by the same author that are cited are discussed in little detail. In addition to their interesting findings, these studies provide a direct measure of susceptibility to motion sickness, which is fundamental to the rational for the current manuscript. I think it is misleading to downplay their role in your manuscript. I would strongly recommend that you include a consideration of these studies the introduction and the discussion sections of your manuscript.

Author Response to ‘’major concern 2”

Thank you for this encouragement to improve the framing and rationale of the paper in light of this literature base. After re-reviewing the work of Drummond, the one with focussed relevance to this paper is certainly the one you suggested—as this is a rare example of causing/exacerbating pain with experimentally induced sensory conflict. This certainly adds weight to the relevance of the study.

In addition to my two major concerns, I have a few minor comments that I believe would improve the manuscript. Below, I have organised these comments by section and line number.

Abstract

Line 22: The “ACOVA suggested”, it did not confirm.

Line 26: Lower than what?

Introduction

I think it is also appropriate to mention that Harris used motion sickness as an analogy when proposing his theory.

Line 64: The efficacy of mirror therapy is overstated. There are several meta-analyses that conclude that the evidence is not sufficiently strong to recommend mirror therapy as a treatment for phantom limb pain (Thieme et al., 2016, The Journal of Pain; Wittkopf et al., 2017, European Journal of Pain). Most of the evidence in favour of mirror therapy for phantom limb pain is based on case studies. I think more detail is needed in this sentence for it to be accurate.

Method

Line 109: How did participants “self-identify” with chronic pain conditions?

I think it is more appropriate to mention that you excluded any participants who had pain below the age of 12 years in the method section rather than leaving it for the discussion (i.e. line 225.

Line 138: Would you not expect pain and medication to induce a recall bias too?

Line 149: There is a reference missing to support the statement about the assumptions of ANOVA.

Results

Line 169: Strictly speaking, the analysis did not find evidence of a difference (i.e. it did not find evidence of no difference). This type of frequentist statistics cannot confirm a null-hypothesis, which would require a test of equivalence (e.g. for a frequentist alternative see Lakens, 2017, Social Psychology and Personality Science).

Line 184: What was the R2 for Gender in your ANCOVA?

Line 188: Greater than what?

Line 196: Figure 1: Not clear what the asterisks refer to. Presumably it is a significance level of the child-adult comparison per group. Either way, this information should be stated explicitly in the caption. It would also be useful to note what the maximum possible score is on the MSSQ-Short.

Discussion

Line 209: I do not think it is accurate to say that this is the first study to propose a sensory processing phenotype. The theory you are testing used motion sickness as an analogy, several previous studies have looked at motion sickness and pain, and there is an entire field dedicated to finding sensory phenotypes of pain (i.e. quantitative sensory testing). So I think some moderation here is warranted.

Line 245: In support of your argument, there is also greater evidence that sensorimotor incongruence is related to pain for people with FMS than for other conditions (see Vittersø et al. 2022, Neuroscience & Biobehavioral Reviews).

Line 286: The review article that you cite (i.e. Vittersø et al. 2022, Neuroscience & Biobehavioral Reviews) contains about 350 references to predominantly empirical studies that are relevant to understanding the sensorimotor theory of pain. So I do not think it is entirely accurate to say that the theory “lacks substantial empirical support”. I would suggest a more nuanced assessment of its empirical backing, although I agree that the evidence base is mixed. 

Comments on the Quality of English Language

Line 230. Typo “.Although” – no space after full stop.

Line 238: Typo. Sentence ends with a quotation mark.

Line 240: Reported not exhibited.

Line 252: Unclear sentence. Are you saying that pre-pain scores indicate a pre-existing predisposition to pain? I think this sentence needs to be simplified.

Line 278: Typo. “Top of form; Bottom of form”.

Submission Date

04 June 2023

Date of this review

06 Jun 2023 12:44:5

Author Response to ‘minor comments’ above

  • Typographical, grammatical and points of clarity have all been addressed, with thanks.
  • Harris’s initial use of motion sickness as an analogy has been acknowledged.
  • While I had only claimed mirror therapy had shown relieving properties in ‘some subgroups of people with PLP’ I have further tempered this claim by removing and replacing this sentence:

Removed section: The principle behind mirror therapy-induced pain relief, that has been shown for subgroups of people with phantom limb pain[10] further supports this theory—since visual feedback provided by the reflected intact limb aligns with the intended movement, thereby restoring sensorimotor congruence.

Replaced by: The principle behind mirror therapy is consistent with this theory, since visual feedback provided by the reflected intact limb aligns with the intended movement, thereby restoring sensorimotor congruence. However, it seems that at best mirror therapy is only effective in subgroups of phantom limb pain[10], with metaanalyses suggesting an overall analgesic effect of just 1.3/10 (on a 10-point Visual Analogue Pain Rating Scale)[11].  

  • I have removed the reference to this being ‘the first’ to propose a sensory processing phenotype. I had meant to say something about the uniqueness of the idea of a specific phenotype linked to processing of incongruent sensory data. However, I can see this was communicated poorly and generally overstated.

Removed section: This study is the first to propose and investigate the hypothesis that a sensory processing-related phenotype may exist, predisposing individuals to both motion sickness susceptibility and chronic pain.

Replaced by: This study was based on the proposition that a sensory processing phenotype may exist relating to the management of conflicting sensorimotor data, and that this may predispose individuals to both motion sickness susceptibility and chronic pain.

Reviewer 2 Report

Dear editors and authors:

The manuscript seems to be of high quality, not only because of the methodology, but also because the topic is relevant to clinical transfer, which is ultimately the objective of the research.

Advances in the neuroscience of the brain is implemented in the clinical performance of physiotherapists, with which the conclusions of this study acquire all the relevance.

Congratulations to the authors.

Author Response

Thank you for this summary and for your time reviewing the manuscript.

Reviewer 3 Report

This retrospective cross-sectional study is the first to correlate motion sickness and certain types of chronic pain. The author relies on the hypothesis that there could be a phenotype related to sensory processing, which predisposes individuals to motion sickness and some types of chronic pain. In this sense, the paper presents the theory of sensorimotor incongruence of chronic pain.  

The paper has all the necessary parts and is well structured. The main limitation of this study was its reliance on retrospective reports, according to the author.

Some suggestions are

- to better explain why the group of patients with rheumatoid arthritis was taken as a control group.

-the presentation of statistical data needs to be completed in the given tables. -there is a discrepancy between the legend and the data shown in Table 2.

- if there is no data on the phenomenon of sensorimotor incongruence for the mentioned painful conditions that correlate with motion sickness, this should be emphasized and stated as a limitation of the interpretation of the results.

Author Response

Thank you for your time reviewing the manuscript.

  • The reasoning for including RA as an additional control group has been expanded as below:

“Rheumatoid arthritis is a persistent pain condition with a clear tissue-based inflammatory mechanism, in contrast to other conditions where local tissue factors are likely less dominant and central nervous system contributors are present. Therefore, rheumatoid arthritis was included as an additional control group to help control for potential confounders, such as current pain and recall bias.” That is, rheumatoid arthritis provides a distinct example of a pain condition driven by tissue inflammation, which may help to isolate the specific effects of other factors under investigation.”

  • Table legends have been corrected.
  • I am not 100% clear on your final point, but I interpret it as follows: You are suggesting that there is no existing data (or even objective way to quantify) whether there is sensorimotor incongruence/conflict present for any of the mentioned painful conditions (or individuals within those conditions). I would tend to agree, and indeed this limits the ability of this idea to move out of the theoretical domain. I have noted something to this effect in the limitations as follows:

It is worth also noting that the idea that sensorimotor conflict may be present in people with persistent pain remains a theoretical idea. Moreover, methods to identify whether someone is in such a state do not exist. This presents a significant limitation to the current study and a challenge to future progress.